# Highly robust supramolecular polymer networks crosslinked by a tiny amount of metallacycles

Lang He [1], Yu Jiang[2], Jialin Wei[1], Zibin Zhang [1], Tao Hong[2], Zhiqiang Ren[3], Jianying Huang [2] ✉, Feihe Huang [4,5] ✉, Peter J. Stang [6] ✉ & Shijun Li [1] ✉

Supramolecular polymeric materials have exhibited attractive features such as self-healing, reversibility, and stimuli-responsiveness. However, on account of the weak bonding nature of most noncovalent interactions, it remains a great challenge to construct supramolecular polymeric materials with high robustness. Moreover, high usage of supramolecular units is usually necessary to promote the formation of robust supramolecular polymeric materials, which restrains their applications. Herein, we describe the construction of highly robust supramolecular polymer networks by using only a tiny amount of metallacycles as the supramolecular crosslinkers. A norbornene ring-opening metathesis copolymer with a 120° dipyridine ligand is prepared and self-assembled with a 60° or 120° Pt(II) acceptor to fabricate the metallacycle-crosslinked polymer networks. With only 0.28 mol% or less pendant dipyridine units to form the metallacycle crosslinkers, the mechanical properties of the polymers are significantly enhanced. The tensile strengths, Young's moduli, and toughness of the reinforced polymers reach up to more than 20 MPa, 600 MPa, and 150 MJ/m$^3$, respectively. Controllable destruction and reconstruction of the metallacycle-crosslinked polymer networks are further demonstrated by the sequential addition of tetrabutylammonium bromide and silver triflate, indicative of good stimuli-responsiveness of the networks. These remarkable performances are attributed to the thermodynamically stable, but dynamic metallacycle-based supramolecular coordination complexes that offer strong linkages with good adaptive characteristics.

Deep cross-fusion of supramolecular chemistry and traditional polymer science has driven the rapid growth of supramolecular polymers[1–7]. Various supramolecular polymers have been constructed by using different recognition motifs, such as hydrogen bonding[8–12], π– π stacking[13–16], host–guest complexations[17–20], ionic attraction[21–23], and metal–ligand interactions[24–26], which allow the corresponding materials to be dynamic, reversible, recoverable, and stimuli-responsive. These distinctive advantages make them highly attractive as promising

[1]College of Material, Chemistry and Chemical Engineering, Key Laboratory of Organosilicon Chemistry and Material Technology of Ministry of Education, Hangzhou Normal University, Hangzhou, P. R. China. [2]College of Food Science and Biotechnology, Zhejiang Gongshang University, Hangzhou, P. R. China. [3]School of Materials Science and Engineering, Peking University, Beijing, P. R. China. [4]Stoddart Institute of Molecular Science, Department of Chemistry, Zhejiang University, Hangzhou, P. R. China. [5]Zhejiang-Israel Joint Laboratory of Self-Assembling Functional Materials, ZJU-Hangzhou Global Scientific and Technological Innovation Center, Zhejiang University, Hangzhou, P. R. China. [6]Department of Chemistry, University of Utah, Salt Lake City, UT, USA. ✉e-mail: huangjy@zjgsu.edu.cn; fhuang@zju.edu.cn; stang@chem.utah.edu; l_shijun@hznu.edu.cn

adaptive materials. However, because of the use of relatively weak non-covalent bonds, supramolecular polymers usually display weak mechanical strength compared with covalent polymers, which inevitably restricts their practical applications[27-29].

To overcome this issue, the integration of covalent polymers and supramolecular polymers could be adopted to fabricate polymeric materials with high robustness[30-33]. Synergistic effects of using covalent and non-covalent systems are beneficial to improve the robustness and meanwhile retain the dynamic characteristics of supramolecular polymers[34-36]. Such polymeric networks crosslinked by supramolecular interactions[37-40], dynamic covalent bonds[41] or mechanically interlocked bonds[42-44] were proven to have intriguing properties that cannot be acquired by covalently crosslinked polymers. Owing to the relative weakness of single supramolecular interactions, multiple and strong interactions are preferable in the crosslinkers. Even so, to attain relatively high robustness, a high percentage of supramolecular units (~10 mol% or more) in the networks was generally necessary to ensure strong crosslinking[45,46]. Nonetheless, high usage of the supramolecular crosslinker units restricts the motion of segments, making the materials more brittle and less stretchable, as well as reducing their practicality. Small usage of supramolecular units is greatly desired to solve this problem.

Over the past decades, coordination-driven self-assembly has attracted much attention, accompanied by the emergence of many supramolecular coordination complexes (SCCs) with well-defined shapes and sizes, including metallacycles and metallacages[47-51]. These SCCs not only have interesting topological architectures but also exhibit extensive applications in many fields[52-56]. Coordination-driven self-assembly possesses great advantages, such as high efficiency, wide diversity, and dynamic and reversible characteristics, which enable it as a very powerful tool in supramolecular self-assembly[57-60]. The unique advantages of coordination-driven self-assembly have also led it to be used for the construction of linear supramolecular polymers from small supramolecular building blocks, which has also proven it is a strong and tolerant dynamic polymerization method[61-64]. Herein, we utilize the self-assembled Pt(II)-based metallacycles to crosslink the linear polymers. With the metallacycles as the strong crosslinkers, we fabricate highly robust ring-opening metathesis polymer networks and unexpectedly find that the presence of only a very small amount of metallacycle crosslinkers is enough to greatly strengthen the mechanical properties of the polymers.

## Results
### Design, synthesis, and structural characterization
A norbornene monomer (M-1) with a 120° dipyridine substituent was first synthesized in six steps in a 45% overall yield from the phthalimide-derived bromide 1 (Fig. 1). The SN2 substitution reaction of 1 with 3,5-dibromophenol produced compound 2 (Supplementary Figs. 1–3). After Sonogashira cross-coupling of 2 with trimethylsilylacetylene and then deprotection of trimethylsilyl (TMS) groups of 3 (Supplementary Figs. 4–6), the intermediate 4 (Supplementary Figs. 7–9) was achieved. The cross-coupling reaction of 4 with 4-bromopyridine provided 5 (Supplementary Figs. 10–12). Hydrazinolysis of 5 generated 6 (Supplementary Figs. 13–15) and the imidization of 6 with *cis*-5-norbornene-*endo*-2,3-dicarboxylic anhydride furnished the monomer M-1 (Supplementary Figs. 16–18). The introduction of the 120° dipyridine ligand on M-1 enables self-assembly to metallacycles.

Two-component coordination-driven self-assembly of the monomer M-1 was subsequently examined. Self-assembly of M-1 with the 60° Pt(II)-based acceptor 7 in 1:1 mole ratio afforded a [2 + 2] Pt(II)-coordinated rhomboid (9), while self-assembly of M-1 with the 120° Pt(II)-based acceptor 8 produced a [3 + 3] hexagonal metallacycle (10), in quantitative yields (Fig. 2). The ³¹P{¹H} and ¹H NMR spectroscopy of the assemblies indicated the formation of single, discrete

species with highly symmetric architectures (Fig. 3). All proton signals were clearly assigned and apparent shifts of the proton peaks were observed in the ¹H NMR spectra of 9 (Fig. 3b and Supplementary Fig. 23) and 10 (Fig. 3e and Supplementary Fig. 27), as compared with those of their corresponding building blocks M-1, 7 and 8 (Supplementary Figs. 16, 19, and 21). The ³¹P{¹H} NMR spectra of both 9 (Fig. 3h and Supplementary Fig. 24) and 10 (Fig. 3j and Supplementary Fig. 28) displayed a single peak flanked by two ¹⁹⁵Pt satellites, consistent with the homologated Pt–N coordination environment. Obvious upfield shifts of the ³¹P{¹H} signals were observed after the formation of the two metallacycles (Fig. 3g–j and Supplementary Figs. 20, 22, 24, 28).

The structures of metallacycles were further proven by electrospray ionization mass spectrometry (ESI-MS). The main peaks in the ESI-MS of 9 and 10 all supported the formation of [2 + 2] rhomboid 9 (Supplementary Fig. 25) and [3 + 3] hexagon 10 (Supplementary Fig. 29). All of the peaks are isotopically resolved and agree very well with their theoretical distributions (Supplementary Figs. 26 and 30). No peaks were found from the assemblies with other stoichiometries. Fortunately, single crystals of the rhombic metallacycle were obtained by vapor diffusion of hexane into a dichloromethane solution of 9 at room temperature. The single crystal X-ray analysis unambiguously confirmed the formation of rhomboid 9 (Fig. 3k, Supplementary Fig. 49, and Supplementary Table 4).

Copolymerization of M-1 and a diester-derived norbornene monomer M-2 was carried out through ring-opening metathesis polymerization (ROMP). It was found that the copolymerization reaction could not be catalyzed by the usually used Grubbs' catalysts, including 1st, 2nd, and 3rd generation, but could be catalyzed by a pyridine-modified 2nd generation Grubbs' catalyst[65]. By using this modified catalyst, the covalent copolymers (CPs) were synthesized from the mixed monomers with a feed ratio of M-1/M-2 = 1:60 (for CP-1), 1:180 (for CP-2), 1:360 (for CP-3), and 1:600 (for CP-4), respectively. The results of gel permeation chromatography (GPC) indicated that the average molecular weight ($M_n$) values of copolymers CP-1 (Supplementary Fig. 32), CP-2 (Supplementary Fig. 34), CP-3 (Supplementary Fig. 36), and CP-4 (Supplementary Fig. 38) were 40 kDa, 47 kDa, 50 kDa, and 64 kDa, respectively, with similar polydispersity index (PDI) ($M_w/M_n$ = 1.45 ~ 1.67) using polystyrene (PS) standards. The characteristic peaks of the dipyridine moieties shown in the ¹H NMR spectra of the copolymers proved that the 120° dipyridine ligand was introduced into the side chains of CPs (Supplementary Figs. 31, 33, 35, and 37 and Supplementary Table 1). Subsequently, coordination-driven self-assemblies of CPs were performed by mixing the CPs with the 60° Pt(II) acceptor 7 and the 120° acceptor 8, respectively, in tetrahydrofuran (THF) solution to generate the corresponding metallacycle-crosslinked polymer networks (MCPNs). After that, the solution viscosity increased significantly (Supplementary Fig. 39), indicative of the formation of crosslinked polymers through the self-assembly of Pt(II) metallacycles among the linear polymers. Due to the strong crosslinking of metallacycles, the reaction mixture of polymer networks (CP-1R, CP-1H, CP-2R, and CP-2H) made from CP-1 (with 1.64 mol% dipyridine units) and CP-2 (with 0.55 mol% dipyridine units) even changed to gels quickly, while the polymer networks (CP-3R, CP-3H, CP-4R, and CP-4H) from CP-3 (with 0.28 mol% dipyridine units) and CP-4 (with 0.17 mol% dipyridine units) remained soluble but with much higher viscosities. Since the MCPNs made from CP-1 and CP-2 solidified into gels, only the MCPNs from CP-3 and CP-4 were further investigated in the following sections. We also attempted an alternative method, polymerization after the formation of metallacycles, but only obtained solutions with low viscosity, indicating that the polymerization was unsuccessful. This is probably because the presence of these metallacycles impedes the catalytic action of the Ru(II)-based catalyst for polymerization.

Thermal properties can reflect the internal structure of polymeric materials. With the MCPNs in hand, their basic thermal properties were

**Fig. 1 | Synthesis of monomer M-1. a** Synthetic route of **M-1. b** Cartoon representation of **M-1.**

investigated by thermal gravimetric analysis (TGA) and differential scanning calorimetry (DSC). The TGA curves demonstrated that the decomposition temperatures (5% weight loss) of the **MCPN**s were all higher than 250 °C, indicative of their good thermal stability (Supplementary Fig. 40). As shown in Fig. 4a, the DSC studies indicated that all the **MCPN**s showed obvious glass transition, probably owing to the relaxation of the polynorbornene segments. Comparatively, all the **MCPN**s possessed higher glass transition temperatures ($T_g$) (42.9 °C for **CP-4H**, 45.3 °C for **CP-4R**, 46.9 °C for **CP-3H**, and 48.1 °C for **CP-3R**) than the linear convent polymer **CP-3** (41.8 °C), indicating that the crosslinking of Pt(II)-based metallacycles restrained the movement of polynorbornene segments. Noticeably, $T_g$ of **CP-3R** and **CP-3H** were higher than those of **CP-4R** and **CP-4H**, respectively, implying that the increase in crosslinking density makes the movement of segments more difficult. In addition, **CP-3R** and **CP-4R** had higher $T_g$ than **CP-3H** and **CP-4H**, respectively, suggesting that the polymers crosslinked by the rhomboid metallacycle are more stable and arranged in better order than those crosslinked by the hexagonal metallacycle. This is probably because the hexagon links to three polymer branches, while the rhomboid links to only two polymer chains. The entanglement of polymer chains makes the [3 + 3] self-assembly more difficult than the [2 + 2] self-assembly.

## Mechanical properties of MCPNs, CP-3, and controls

To manifest the performance of Pt(II)-based metallacycles as crosslinkers, the mechanical properties of **MCPN**s and **CP-3** were systematically investigated through tensile tests. The stress–strain curves, the calculated Young's moduli, and the toughness of **MCPN**s and **CP-3** are illustrated in Fig. 4b, c. Both the **MCPN**s and **CP-3** exhibited yield

behavior because all of them were in the glassy state at room temperature. The yield strengths of all **MCPN**s exceeded 20 MPa, significantly higher than that of **CP-3** (about 5 MPa, Supplementary Table 2). The tensile strengths and Young's moduli of **MCPN**s exceeded 15 MPa and 350 MPa, respectively, also much higher than those of **CP-3**. The toughness of **CP-3R** and **CP-3H** also remarkably increased from less than 50 MJ/m³ for **CP-3** to more than 150 MJ/m³. These results revealed that the mechanical properties of the polymers were greatly improved after crosslinking by the metallacycles, despite the fact that only tiny amounts of crosslinkers were used. Overall, **CP-3R** and **CP-3H** possessed more balanced and comprehensive mechanical performance in comparison with **CP-4R** and **CP-4H**, so they were selected as representative samples for subsequent investigation. Although their toughness was almost the same, the Young's modulus of **CP-3R** was higher than that of **CP-3H**. The higher Young's modulus can reflect the better stability of the rhomboid metallacycle crosslinked network, which is consistent with the results of the DSC discussed above. This is probably because the self-assembly of [2 + 2] rhombic metallacycles involves fewer components, while self-assembly of the [3 + 3] hexagonal metallacycles involves more components and thus is relatively slower and more difficult in the polymer solutions. Therefore, more incomplete assemblies (presented in Supplementary Figs. 41 and 42) may exist in **CP-3H** than in **CP-3R**. Although some defective assemblies can also act as crosslinkers, they are single-coordinated and weaker than the double-coordinated metallacycles. In other words, the rhomboid-crosslinked polymer networks have relatively fewer soft polymer chains not participating in the self-assembly. For the same reason, the **MCPN**s crosslinked by the rhomboid metallacycle would result in a more ordered arrangement of the polymer chains, also

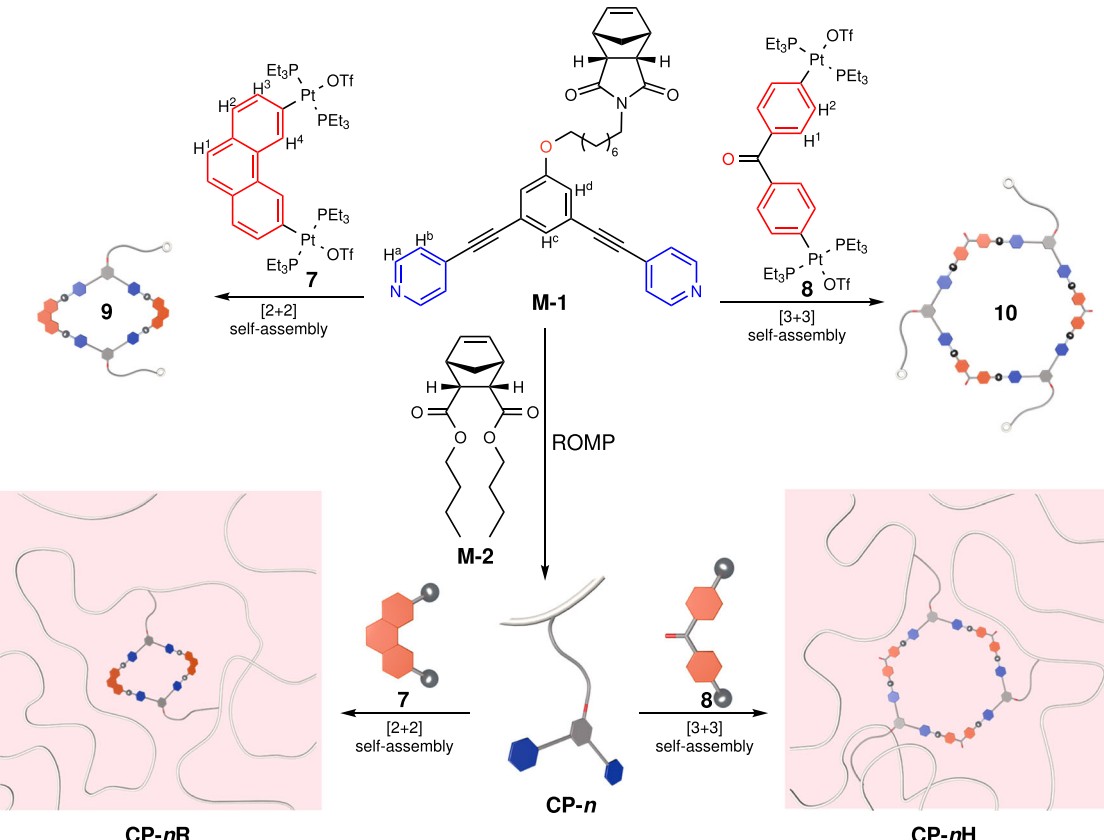

**Fig. 2 | Preparation of the metallacycles and metallacycle-crosslinked polymer networks.** Ring-opening metathesis polymerization (ROMP) reaction of the monomer **M-1** and **M-2** is catalyzed by a pyridine-modified Grubbs II catalyst to produce the polymers **CP-n**. The cartoon structures represent the rhombic and hexagonal metallacycles and their building blocks. Color code: red = Pt(II) acceptor; blue = dipyridine ligand.

leading to the higher modulus of **CP-3R**. Meanwhile, because the partially broken hexagons can still act as crosslinkers after one of the branch chains on the hexagonal metallacycles is broken off, **CP-3H** has a higher elongation at break.

To better understand the role of the supramolecular Pt(II)-based metallacycle crosslinkers in enhancing mechanical performance, two controls, <sup>con</sup>**CP-3R** and <sup>con</sup>**CP-3H** were designed and synthesized. Their components are similar to **CP-3R** and **CP-3H**, respectively, except that the platinum of the control samples is in the form of a dibromide acceptor which cannot coordinate with the dipyridine ligand (Supplementary Fig. 43). The mechanical performances of the two control were measured by tensile tests and compared with those of **CP-3R** and **CP-3H** (Fig. 4d, e). The results indicated that <sup>con</sup>**CP-3R** and <sup>con</sup>**CP-3H** had mechanical properties similar to **CP-3**, while Young's modulus, tensile strength, and toughness of the control samples were much lower than those of **CP-3R** and **CP-3H**, respectively. Such significant differences in performance indicated that the crosslinking of polymer chains by the Pt(II)-based metallacycles plays a crucially important role in the enhancement of mechanical properties.

The cyclic tensile tests at a maximum strain of 500% demonstrated that **CP-3R** and **CP-3H** possessed higher energy dissipation capacity than the two controls (Fig. 4f). During the stretching process, polymer networks of **CP-3R** and **CP-3H** dissociated under external forces until the crosslinking points of the metallacycles were destroyed, which can dissipate a large amount of energy. The intrinsic dynamic features of coordination-driven self-assembly endow the metallacycles with reversible destruction/reformation, which greatly benefits energy dissipation (Fig. 5). The strong coordination interaction and good stability of the metallacycles largely increased the tolerance to an external force. Therefore, **CP-3R** and **CP-3H** possess outstanding energy dissipation capability.

**Insights into the mechanical performance of CP-3R and CP-3H**
As shown by the tensile experiments in the previous section, the mechanical performances of **MCPN**s underwent extraordinary improvement due to the introduction of robust and meanwhile dynamic metallacycle-based supramolecular crosslinking. It was speculated that the enhanced tensile strength, Young's moduli, and toughness of **MCPN**s originated from the supramolecular networks formed during the assembly process. In order to have an in-depth understanding of the improvement in mechanical properties, dynamic mechanical analysis (DMA) experiments were conducted on **CP-3**, **CP-3R**, and **CP-3H** (Fig. 6a–c). For the linear polymer **CP-3**, the storage modulus (G′) began to decrease as the temperature increased. When it was heated to 30 °C, G′ decreased significantly and then reached a value close to the loss modulus (G″) at about 48 °C, which indicated that the polymer gradually transformed to a rubbery state from a glassy state. During the heating process, **CP-3R** and **CP-3H** exhibited dynamic thermal mechanical behaviors similar to **CP-3**. The difference is that both **CP-3R** and **CP-3H** have a storage modulus of over 400 MPa before 32 °C, attributed to the robust metallacyclic crosslinking points in the networks of **CP-3R** and **CP-3H**. It is worth noting that **CP-3R** has a higher storage modulus than **CP-3H**, indicating that **CP-3R** has a more stable crosslinking entanglement network (Supplementary Fig. 44), which is consistent with the results of DSC and tensile tests. The value of G′ began to decrease sharply upon further heating, indicative of the destruction of the Pt(II)-coordinated metallacycles which caused rapid collapse of the crosslinked polymer networks. In addition, there is only a slight difference in the value of tanδ among **CP-3**, **CP-3R**, and

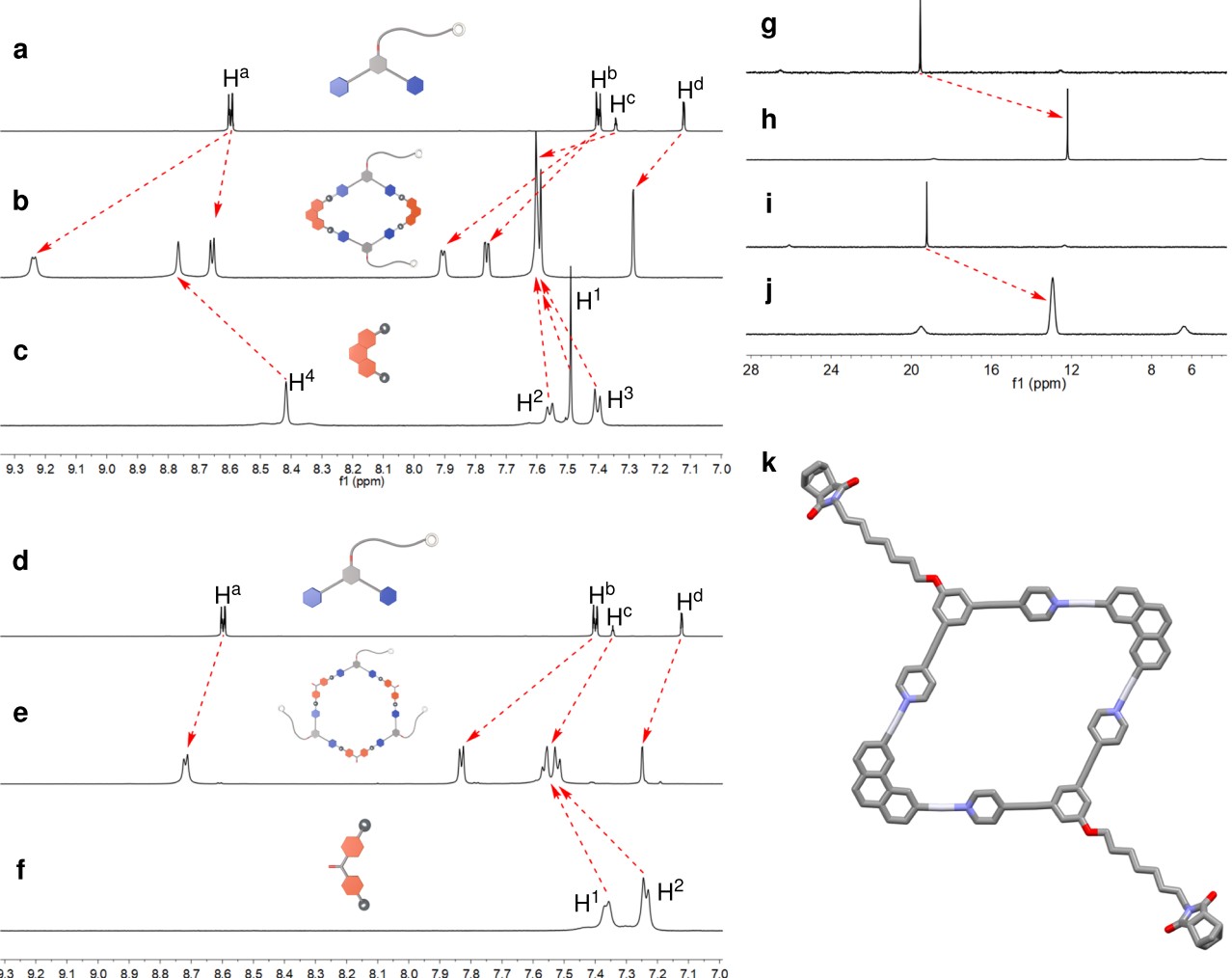

**Fig. 3 | Structural characterization of metallacycles.** Partial $^1$H NMR spectra (500 MHz, CD$_2$Cl$_2$, 298 K) of **M-1** (**a**), metallacycle **9** (**b**), **7** (**c**), **M-1** (**d**), metallacycle **10** (**e**) and **8** (**f**). $^{31}$P{$^1$H} spectra (202 MHz, CD$_2$Cl$_2$, 298 K) of **7** (**g**), **9** (**h**), **8** (**i**) and **10** (**j**). **k** X-ray crystal structure of metallacycle **9**. Counterions, solvents, H atoms, and triethylphosphine ligands on the platinum atoms have been omitted for clarity.

**CP-3H**, suggesting that they have similar aggregated structures after the metallacycles are disassembled.

To further investigate the enhancement of mechanical properties derived from the Pt(II)-coordinated metallacycles, scanning electron micrography (SEM) characterization on the fracture surface was performed (Fig. 6d–f). The cryo-fractured surface of **CP-3** is relatively smooth, while those of **CP-3R** and **CP-3H** are relatively rough, which can be attributed to the destruction of strong metallacycle networks distributed inside the polymers during the process of fracture. The network gels were also characterized by SEM. It was found that the gel film of **CP-3** could be dissolved in 1,2-dichloroethane, but the gel films of **CP-3R** and **CP-3H** prepared in situ from their precursors could only be swollen (Supplementary Fig. 45). The swollen films of **CP-3R** and **CP-3H** were changed to xerogels by freeze-drying. SEM images of the xerogels of **CP-3R** and **CP-3H** showed crosslinked network structures, clearly indicating the formation of crosslinked polymer networks (Supplementary Figs. 46 and 47).

**Dynamic properties of CP-3R and CP-3H**

The studies in the previous sections demonstrated that the robust metallacycle-based crosslinking is capable of effectively stabilizing the polymer networks to withstand destruction under an external force, thus exhibiting increased strength and toughness of the **MCPN**s.

Meanwhile, the inherent dynamic features of coordination-based supramolecular assembly can endow the **MCPN**s with attractive dynamic properties. Therefore, we subsequently investigated the dynamic behavior of **CP-3R** and **CP-3H**.

First, the energy dissipation capabilities of **CP-3R** and **CP-3H** were assessed through cyclic tensile tests at different maximum strains ranging from 100% to 500%. As shown in Fig. 7a, b, when the strain was increased gradually, **CP-3R** and **CP-3H** presented large hysteresis loops accompanied by higher residual strains. After the energy dissipations were calculated by using the hysteresis areas, the relationships between the energy dissipation values and the strains were found to be approximately linear (Fig. 7c). It was demonstrated that the damping capacities of **CP-3R** and **CP-3H** kept a steadily high value of about 80% at high strains, indicative of their good ability in energy dissipation (Fig. 7c). These results revealed that a small amount of dynamic Pt(II)-based metallacycles was able to provide sacrificial bonds to effectively dissipate the input energy. As it can be seen from the DSC (Fig. 4a) and DMA (Fig. 6b, c) results, the glass transition temperature of our norbornene copolymer is slightly higher than room temperature. Therefore, at room temperature, the polymer segments are in a frozen state. During external stretching, the frozen segments would be unfrozen and undergo movement. Although the Pt(II) metallacycles have been introduced as the crosslinkers, the amount of metallacycles is very

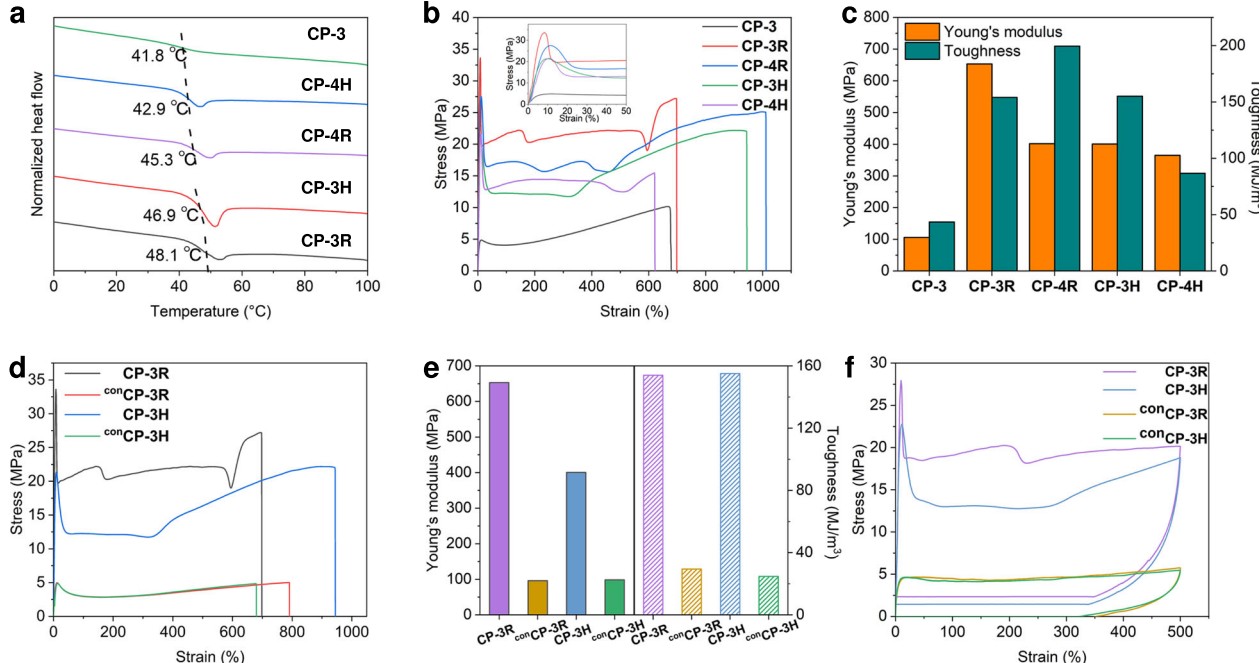

**Fig. 4 | Fundamental performance characterizations of MCPNs. a** DSC curve of **MCPN**s and **CP-3** recorded in the first heating scan from 0 to 100 °C with a heating rate of 20 °C/min. **b** Stress–strain curves of **MCPN**s and **CP-3** recorded with a deformation rate of 100 mm/min. **c** Young's moduli and toughness of **MCPN**s and **CP-3** calculated from their stress–strain curves. **d** Stress–strain curves of **CP-3R**, **CP-3H**, $^{con}$**CP-3R**, and $^{con}$**CP-3H** were recorded with a deformation rate of 100 mm/min. **e** Young's moduli and toughness of **CP-3R**, **CP-3H**, $^{con}$**CP-3R**, and $^{con}$**CP-3H** calculated from their stress–strain curves. **f** Cyclic tensile test curves of **CP-3R**, **CP-3H**, $^{con}$**CP-3R**, and $^{con}$**CP-3H** loaded at a strain of 500% under the deformation rate of 100 mm/min.

small, so the polymer segments can move under external forces. After unloading the external forces, the movement of segments needs to overcome the influence of crosslinking and the rigidity of its own segments. During this process, a large amount of mechanical energy was dissipated in the form of heat.

## Stimuli-responsive properties of CP-3R and CP-3H

Stimuli-responsiveness has long been a fascinating feature of supramolecular materials. As only a tiny amount of Pt(II)-based metallacycles were used as the dynamic crosslinkers, we were curious about whether the metallacycle-crosslinked polymers **CP-3R** and **CP-3H** would exhibit stimuli-responsiveness and reveal changeable macroscopic properties under external stimuli. It is known that the Pt−N coordination bonds in the Pt(II)-based metallacycles are easy to be destroyed by introducing competitive bromide anions[66]. When tetrabutylammonium bromide (TBABr) in an equimolar ratio with the platinum atoms was added to the crosslinked polymer solid sample suspended in THF, a solution formed because the introduction of bromide anions disrupts the coordination between platinum and pyridine (Supplementary Fig. 48b, e). After that, equimolar silver trifluoromethylsulfonate (AgOTf) was added. It was found that the viscosity of the solution increased and the solution gelled (Supplementary Fig. 48c, f), because the coordination bonds were restored after removal of bromide anions in the form of silver bromide precipitated from solutions. These studies indicate that the Pt(II)-based metallacycles demonstrate the reversible transformation in response to added ions (Fig. 7d).

The mechanical properties also reveal these reversible transformations. After the crosslinked polymer samples were treated with TBABr, mechanical performances decreased; tensile strength (12.13 ± 0.43 MPa for **CP-3R** + **Br**⁻ and 11.86 ± 0.72 MPa for **CP-3H** + **Br**⁻), Young's moduli (136.17 ± 3.32 MPa for **CP-3R** + **Br**⁻ and 135.66 ± 5.77 MPa for **CP-3H** + **Br**⁻), and toughness (97.14 ± 3.77 MJ/m³ for **CP-3R** + **Br**⁻ and 100.89 ± 6.88 MJ/m³ for **CP-3H** + **Br**⁻) were

observed in comparison with the virgin ones (Fig. 7e–g and Supplementary Table 3). Again the addition of TBABr destroyed the Pt-N coordination bonds, and thus led to the partial decomposition of the **MCPN**s and the dramatic decrease in mechanical properties. The corresponding tensile strength and Young's moduli were partially restored after the subsequent introduction of AgOTf into the solutions. The tensile strengths of **CP-3R** + **Br**⁻ + **Ag**⁺ and **CP-3H** + **Br**⁻ + **Ag**⁺ were restored to 15.74 ± 3.14 MPa and 13.11 ± 2.72 MPa, respectively. The Young's moduli were also significantly recovered to be 407.79 ± 4.72 MPa for **CP-3R** + **Br**⁻ + **Ag**⁺ and 290.9 ± 7.25 MPa for **CP-3H** + **Br**⁻ + **Ag**⁺. However, their elongation at break and toughness were lower than those of **CP-3R** + **Br**⁻ and **CP-3H** + **Br**⁻ before the addition of AgOTf, perhaps due to the formation of insoluble silver bromide. These results revealed that the Pt(II) metallacycle-based supramolecular crosslinking is subject to modulation by external stimuli, thereby rendering the **MCPN**s with stimuli-responsiveness.

## Discussion

In this work, we have presented the design and construction of tough supramolecular polymer networks crosslinked by a tiny amount of metallacycles. A norbornene substituted with a 120° dipyridine ligand was synthesized and copolymerized with a diester-derived norbornene monomer through ROMP to fabricate the corresponding CPs. The copolymers were then self-assembled with 60° or 120° Pt(II) acceptors to construct metallacycle-crosslinked polymer networks. Although only 0.28 mol% or less pendant dipyridine units were used to form the metallacycle crosslinkers, the mechanical properties of the polymers were significantly improved. After crosslinking, the yield strength, tensile strength, Young's modulus, and toughness all increased to several times of the original ones. The profound effect of metallacycle crosslinking on the mechanical properties of the polymers is supposed to benefit from the robust, dynamic, and damage-tolerable features of Pt(II)-coordinated metallacycles. Good stimuli-responsiveness of the metallacycle-crosslinked polymer networks was demonstrated by the

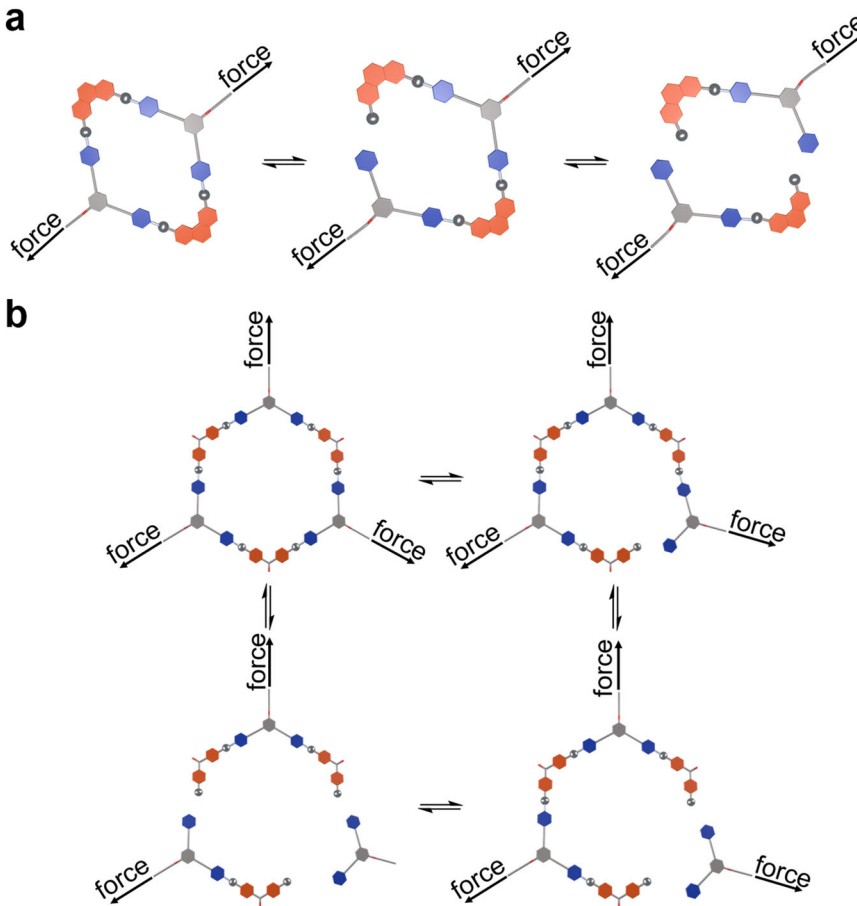

**Fig. 5 | Cartoon representations for destruction and reformation of the metallacycles. a** For rhombic metallacycle. **b** For hexagonal metallacycle.

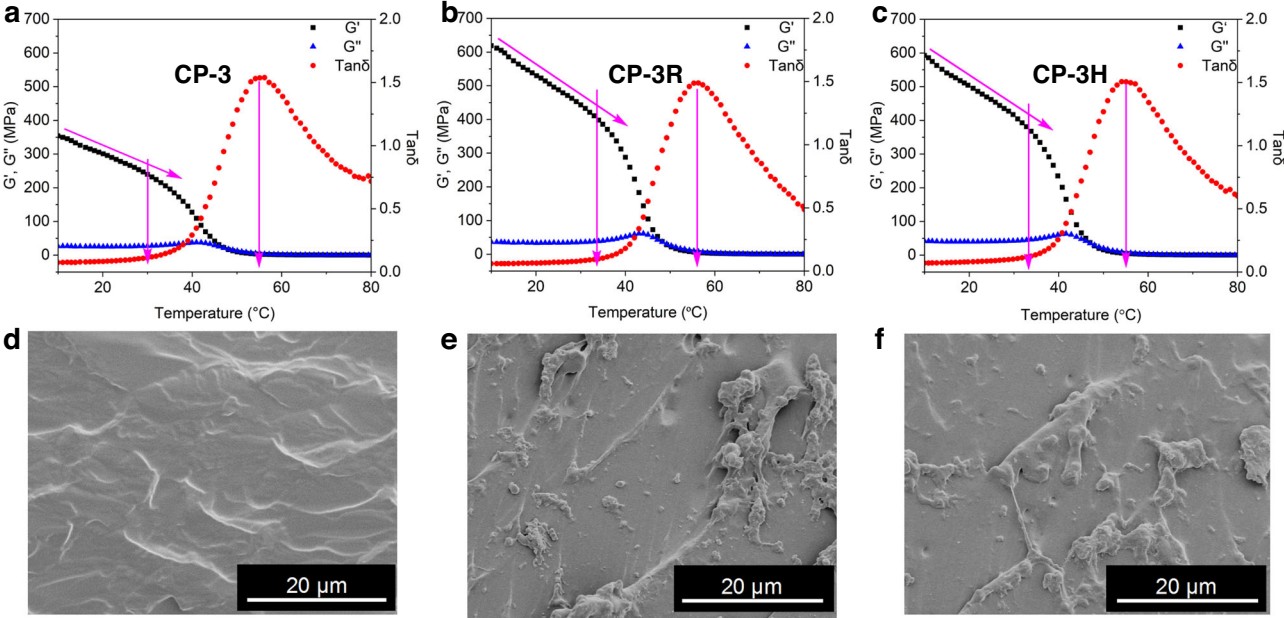

**Fig. 6 | Dynamic thermomechanical properties and micromorphology in the solid state of MCPNs.** DMA curves of **CP-3** (**a**), **CP-3R** (**b**) and **CP-3H** (**c**) were recorded in the heating scan from 10 to 80 °C with a heating rate of 2 °C/min. SEM micrographs of cryofractured surfaces of **CP-3** (**d**), **CP-3R** (**e**) and **CP-3H** (**f**).

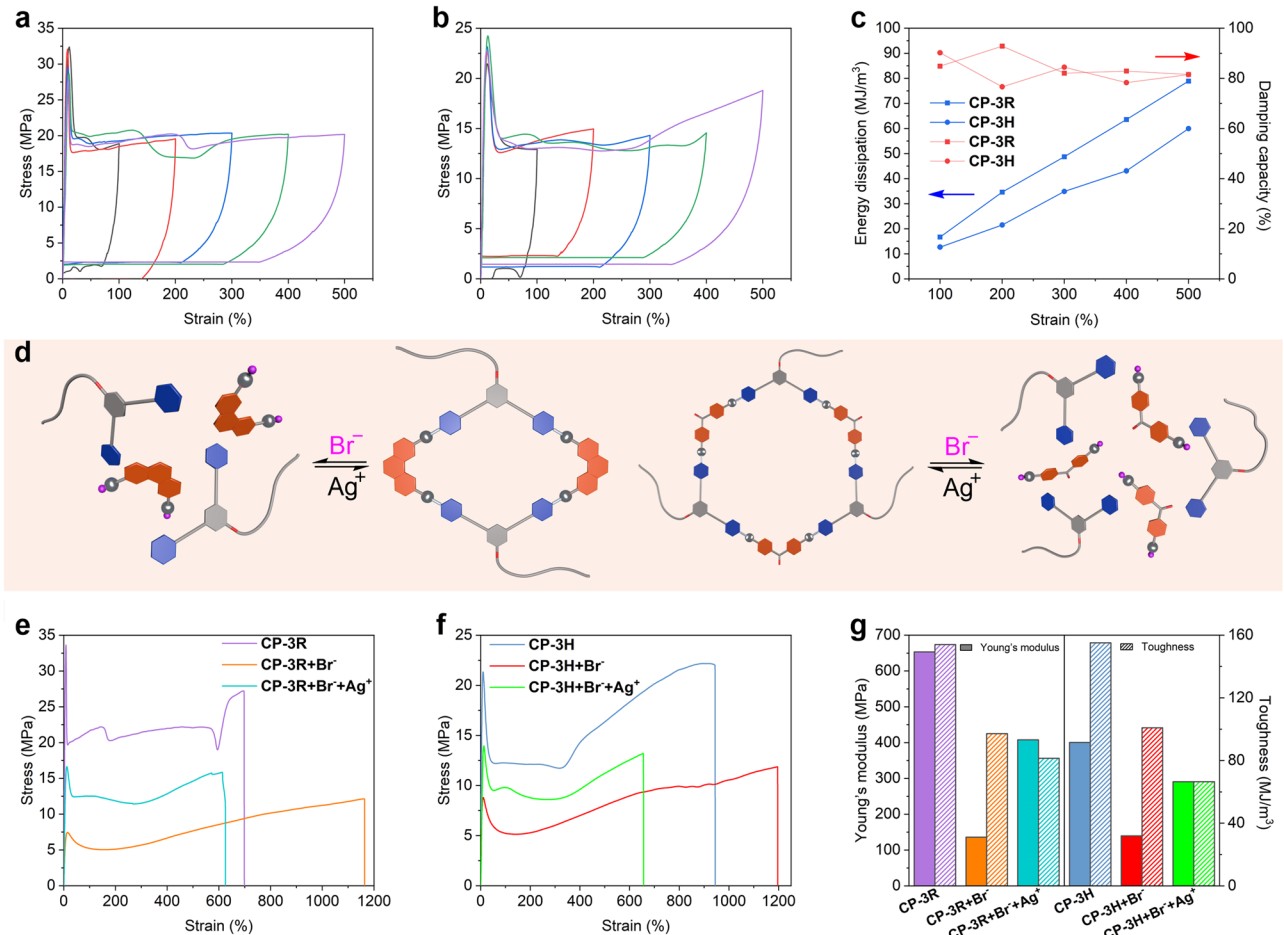

**Fig. 7 | Dynamic and stimuli-responsive properties of CP-3R and CP-3H.** Cyclic tensile test curves of **CP-3R** (**a**) and **CP-3H** (**b**) were recorded with increased maximum strains. **c** Energy dissipation and damping capacity for each circle of the cyclic tensile test curves. **d** Schematic diagram of reversible stimulus-response of rhomboic (left) and hexagonal (right) metallacycle. **e** Stress−strain curves of **CP-3R**, **CP-3R** + **Br⁻**, and **CP-3R** + **Br⁻** + **Ag⁺** were recorded with a deformation rate of 100 mm/min. **f** Stress−strain curves of **CP-3H**, **CP-3H** + **Br⁻**, and **CP-3H** + **Br⁻** + **Ag⁺** were recorded with a deformation rate of 100 mm/min. **g** Young's moduli and toughness of **CP-3R**, **CP-3R** + **Br⁻**, **CP-3R** + **Br⁻** + **Ag⁺**, **CP-3H**, **CP-3H** + **Br⁻**, and **CP-3H** + **Br⁻** + **Ag⁺** calculated from their stress−strain curves.

sequential addition of TBABr and silver triflate. This work provides a strategy to significantly reinforce the mechanical strength of supramolecular polymeric materials by using very small amounts of supramolecular units, meanwhile maintaining their stimuli-responsive features.

## Methods

The synthesis and characterization of compounds presented in this work, the experimental details and additional data of tests are described in the Supplementary Information.

## Data availability

The authors declare that the data supporting the findings of this study are available within the paper and its Supplementary Information file. The X-ray crystallographic data reported in this study are deposited at the Cambridge Crystallographic Data Center (CCDC), under deposition number CCDC 2297067 (compound **9**). The data can be obtained free of charge from CCDC via https://www.ccdc.cam.ac.uk/structures/. All other data are available from the corresponding authors upon reasonable request. Source data are provided with this paper.

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

## Acknowledgements
S.L. thanks the National Natural Science Foundation of China (22071040) and the Natural Science Foundation of Zhejiang Province (LZ24B020005) for financial support, and we thank Jiyong Liu (Department of Chemistry, Zhejiang University, Hangzhou 310058, P. R. China) for the single crystal characterization of metallacycle **9**. F.H. thanks the National Key Research and Development Program of China (2021YFA0910100), the National Natural Science Foundation of China (22035006, 22320102001, and 22350007), the Zhejiang Provincial Natural Science Foundation of China (LD21B020001), the Starry Night Science Fund of Zhejiang University Shanghai Institute for Advanced Study (SN-ZJU-SIAS-006), and the Leading Innovation Team grant from the Department of Science and Technology of Zhejiang Province (2022R01005) for financial support. F.H. thanks the Chemistry Instrumentation Center of Zhejiang University for the technical support.

## Author contributions
J.H., F.H., P.J.S. and S.L. conceived and designed the experiments. L.H., Y.J., J.W. and T.H. performed the synthesis, self-assembly, and characterization experiments. L.H., Z.Z., Z.R., J.H. and S.L. analyzed the data. L.H., F.H., P.J.S. and S.L. wrote the manuscript together.

## Competing interests
The authors declare no competing interests.
