## [Peer Review File · Nature Communications]

Highly robust supramolecular polymer networks crosslinked by a tiny amount of metallacyclesReviewers' Comments:

Reviewer #1:

Remarks to the Author:

Stang et al.

NCOMMS-24-0066

This manuscript describes the design and syntheses of norbornene-based copolymers with small percentages of a pendant bipyridine unit, the complexation of these copolymers with Pt(II) to form metallacyclic crosslinks from the bipyridine pendant groups, and mechanical property tests to demonstrate their improvement via the metallacyclic crosslinks. The impressive results confirm that this constitutes a new approach to supramolecular crosslinking of polymers for property enhancement. Therefore, this paper will be of broad interest to supramolecular chemists, polymer scientists and material scientists; therefore, after the minor issues below have been adequately addressed, it should be published in Nature Communications.

- The pages of the ms and the SI should be numbered!! The SI needs a Table of Contents.
- Page 2: The authors use the term "strong enough". What is "strong enough" in terms of parameters they study? If this question cannot be answered satisfactorily, the term should be removed.
- It is suggested that in the text a table is used to indicate the mol % of dipyriddy monomer M1 used in the polymerization be indicated: 1.7, 0.55, 0.28 and 1.7% for CP-1, CP-2, CP-3 and CP-4. The Table could also contain the average molecular weights and polydispersities.
- Page 5: What is meant by the term "trammel"???
- Reference list: The use of "et al." in the list of author names is restricted by most journals; include the names of the coauthors!
- SI, pages 19-22: There is confusion about the structure numbers.
- SI ¹H NMR spectra, throughout: Unless special conditions are imposed during collection of the spectra, coupling constants can only be estimated to whole number precision. Thus, not $J = 5.4$, 3.0 Hz, but rather $J = 5$, 3 Hz.

These and language issues are noted in the "Track Changes" Word documents attached.

Reviewer #2:

Remarks to the Author:

In this manuscript, Li and coworkers described the construction of highly robust supramolecular polymer networks by using metallacycles as the supramolecular crosslinkers. It should be noted that construction of supramolecular polymeric materials form dynamic metallacycles and covalent polymers has evolved to be one of the most attractive topics within supramolecular chemistry and materials sciences. In this work, Li and coworkers presented the design and construction of tough supramolecular polymer networks crosslinked by a tiny amount of metallacycles. Importantly, after crosslinking, the yield strength, tensile strength, Young's modulus, and toughness could all increase to several times of the original ones. In addition, the obtained gel shows a multiple stimuli responsiveness behavior. The manuscript, with good novelty and scientific value and the data collected are interesting and the overall findings merit publication. Thus, I would like to recommend to be published in Nature Communications after some revisions.

1. Crosslinking process usually produces some intriguing morphologies. Therefore, the morphologies of the supramolecular gel are recommended to be characterized by the microscopy techniques (e.g., SEM, TEM, AFM and so on), which will be helpful to understand the nanostructures of such soft materials.

2. The authors mentioned that the polymers crosslinked by the rhomboid metallacycle are more stable and arrange in better order than those crosslinked by the hexagonal metallacycle. How about the differences of crosslinking degree in supramolecular gel? The referee is also very interested in the underlying mechanism.

3. Why was the pre-synthesis of the block copolymer formation route chosen, rather than polymerization after metallacycle? Did the polymerization interfere with metallacycle disassembly? Perhaps a few comments on the rationale for this strategy could be added, especially any information describing if/why the alternative was not attempted or successful.

We appreciate the reviewers' comments which have greatly improved our manuscript. The specific changes to the manuscript as per request are listed as following.

Response to Reviewer 1:

This manuscript describes the design and syntheses of norbornene-based copolymers with small percentages of a pendant bipyridine unit, the complexation of these copolymers with Pt(II) to form metallacyclic crosslinks from the bipyridine pendant groups, and mechanical property tests to demonstrate their improvement via the metallacyclic crosslinks. The impressive results confirm that this constitutes a new approach to supramolecular crosslinking of polymers for property enhancement. Therefore, this paper will be of broad interest to supramolecular chemists, polymer scientists and material scientists; therefore, after the minor issues below have been adequately addressed, it should be published in Nature Communications.

Reply: We appreciate the reviewer's positive comments on the work in this manuscript.

The pages of the MS and the SI should be numbered!! The SI needs a Table of Contents.

Reply: Thanks. We have numbered the pages of the main text and the Supplementary Information, and added a Table of Contents in the Supplementary Information.

Page 2: The authors use the term "strong enough". What is "strong enough" in terms of parameters they study? If this question cannot be answered satisfactorily, the term should be removed.

Reply: As the reviewer pointed out, it is inappropriate to use the term "strong enough". We have changed it to "strong" on Page 2 in the main text.

It is suggested that in the text a table is used to indicate the mol % of dipyriddy monomer M1 used in the polymerization be indicated: 1.7, 0.55, 0.28 and 1.7% for CP-1, CP-2, CP-3 and CP-4. The Table could also contain the average molecular weights and polydispersities.

Reply: As the reviewer suggested, we have added a table that contains average molecular weights, polydispersities and mol% of dipyriddy monomer **M1** used in the polymerization for **CP-1**, **CP-2**, **CP-3** and **CP-4** on Page S36 in the Supplementary Information.

Page 5: What is meant by the term "trammel"???

Reply: The term "trammel" is not suitable for use here, and we have changed it to "entanglement of polymer chains" on Page 5 in the main text.

Reference list: The use of “et al.” in the list of author names is restricted by most journals; include the names of the coauthors!

Reply: For references, *Nature Communications* uses standard *Nature* referencing style. All authors should be included in reference lists unless there are six or more, in which case only the first author should be given, followed by ‘et al.’. It can be found in the link ‘<https://www.nature.com/ncomms/submit/how-to-submit>’.

SI, pages 19-22: There is confusion about the structure numbers.

Reply: The typing errors of the structure numbers have corrected on Pages S23-25 of the Supplementary Information.

SI ¹H NMR spectra, throughout: Unless special conditions are imposed during collection of the spectra, coupling constants can only be estimated to whole number precision. Thus, not $J = 5.4, 3.0$ Hz, but rather $J = 5, 3$ Hz.

Reply: The coupling constants have changed to be whole number precision in the Supplementary Information.

Response to Reviewer 2

In this manuscript, Li and coworkers described the construction of highly robust supramolecular polymer networks by using metallacycles as the supramolecular crosslinkers. It should be noted that construction of supramolecular polymeric materials form dynamic metallacycles and covalent polymers has evolved to be one of the most attractive topics within supramolecular chemistry and materials sciences. In this work, Li and coworkers presented the design and construction of tough supramolecular polymer networks crosslinked by a tiny amount of metallacycles. Importantly, after crosslinking, the yield strength, tensile strength, Young’s modulus, and toughness could all increase to several times of the original ones. In addition, the obtained gel shows a multiple stimuli responsiveness behavior. The manuscript, with good novelty and scientific value and the data collected are interesting and the overall findings merit publication Thus, I would like to recommend to be published in Nature Communications after some revisions.

Reply: We appreciate the reviewer’s positive comments on the work in this manuscript.

Crosslinking process usually produces some intriguing morphologies. Therefore, the morphologies of the supramolecular gel are recommended to be characterized by the microscopy techniques (e.g., SEM, TEM, AFM and so on), which will be helpful to understand the nanostructures of such soft materials.

Reply: As the reviewer suggested, we have further characterized the gels of **CP-3R** and **CP-3H** by SEM. The gel films of **CP-3R** and **CP-3H** prepared in situ from their precursors were swollen in 1,2-dichloroethane (Supplementary Fig. 45) and then changed to xerogels by freeze-drying. SEM images of the xerogels of **CP-3R** and **CP-3H** clearly showed crosslinked network structures, indicating the formation of crosslinked polymer networks (Supplementary Figs. 46 and 47). The corresponding results and discussions have been added in the main text on Page 8 and in Supplementary Information on Page S45. Moreover, we found that the gels of **CP-3R** and **CP-3H** could not be characterized by TEM because they cannot be dissolved in solvents.

The authors mentioned that the polymers crosslinked by the rhomboid metallacycle are more stable and arrange in better order than those crosslinked by the hexagonal metallacycle. How about the differences of crosslinking degree in supramolecular gel? The referee is also very interested in the underlying mechanism.

Reply: As we found, **CP-3R** has better mechanical properties than **CP-3H**, indicating that **CP-3R** has a more stable crosslinking entanglement network. The glass transition temperature reflects the thermal stability and crosslinking degree of polymer networks. For polymeric materials with similar chemical compositions, **CP-3R** exhibited higher T_g than **CP-3H**, indicating higher stability of polymer networks. And the results from tensile tests showed that **CP-3R** had a higher value of Young's modulus than **CP-3H**, which can also reflect the better stability of the rhombic metallacycle crosslinked network. Storage modulus (G') can reflect the strengths of the entanglement networks in polymeric materials to resist cyclic stress, which can further reflect the crosslinking degree of polymer chains. From the result of that **CP-3R** had higher storage modulus than **CP-3H**, we can know that the crosslinking degree of **CP-3R** is higher than that of **CP-3H**.

The reasons can be speculated from their difference in structures. The self-assembly of [2 + 2] rhombic metallacycles involves less components, while self-assembly of the [3 + 3] hexagonal metallacycles involves more components and thus is relatively slower and more difficult in the polymer solutions. Therefore, more incomplete assemblies (presented in Supplementary Figs. 41 and 42) may exist in **CP-3H** than in **CP-3R**. Although some defective assemblies can also act as crosslinkers, they are single-coordinated and weaker than the double-coordinated metallacycles. In other words, the rhomboid-crosslinked polymer networks have relatively fewer soft polymer chains not participating in the self-assembly. That means **CP-3R** has more effective crosslinking points than **CP-3H**. For the same reason, the MCPNs crosslinked by the rhomboid metallacycle would result in a more ordered arrangement of the polymer chains, which can also lead to higher glass transition temperature and higher modulus. The corresponding discussion has been added on Page 8 in the main text and Page 43 in Supplementary Information. The corresponding discussion have been added in the main text on Pages 6, 7, 8 and in Supplementary Information on Pages 40, 43.

Why was the pre-synthesis of the block copolymer formation route chosen, rather than polymerization after metallacycle? Did the polymerization interfere with metallacycle disassembly? Perhaps a few comments on the rationale for this strategy could be added, especially any information describing if/why the alternative was not attempted or successful.

Reply: We have attempted post-self-assembly method, polymerization after the formation of metallacycles, but only obtained solutions with low viscosity, indicating that the polymerization was unsuccessful. This is probably because the presence of these metallacycles impedes the action of the Ru(II)-based catalyst for polymerization. The corresponding discussion has been added in the main text on Page 5.

Again, we greatly appreciate the reviewers' comments which have greatly improved the quality of our manuscript. With these changes and responses, we hope the revised manuscript is now acceptable for publication in Nature Communications.

Best regards,

Shijun Li

Report ref 2:

This is a revised manuscript resubmitted by Li and coworkers, in which they presented the the construction of highly robust supramolecular polymer networks by using metallacycles as the supramolecular crosslinkers. The authors have revised the draft according to the reviewers' suggestions. I can tell that the quality of the updated version has been dramatically improved. I strongly recommend it to be accpeted by Nature Communications just as it.